# An Assessment of the Pathological Classification and Postoperative Outcome of Focal Cortical Dysplasia by Simultaneous Hybrid PET/MRI

**DOI:** 10.3390/brainsci13040611

**Published:** 2023-04-04

**Authors:** Ning Wang, Lingjie Wang, Yixing Yu, Guangzheng Li, Changhao Cao, Rui Xu, Bin Jiang, Yongfeng Bi, Minjia Xie, Chunhong Hu, Wei Gao, Mo Zhu

**Affiliations:** 1Department of Radiology, The First Affiliated Hospital of Soochow University, Suzhou 215008, China; 2Department of Neurosurgery, The First Affiliated Hospital of Soochow University, Suzhou 215008, China

**Keywords:** focal cortical dysplasia, MRI, hybrid (18F)-FDG PET/MRI, histopathological result, surgical outcome

## Abstract

Objectives: The purpose of this research was to investigate whether MRI and Simultaneous Hybrid PET/MRI images were consistent in the histological classification of patients with focal cortical dysplasia. Additionally, this research aimed to evaluate the postoperative outcomes with the MRI and Simultaneous Hybrid PET/MRI images of focal cortical dysplasia. Methods: A total of 69 cases in this research were evaluated preoperatively for drug-resistant seizures, and then surgical resection procedures of the epileptogenic foci were performed. The postoperative result was histopathologically confirmed as focal cortical dysplasia, and patients then underwent PET and MRI imaging within one month of the seizure. In this study, head MRI was performed using a 3.0 T magnetic resonance scanner (Philips) to obtain 3D T1WI images. The Siemens Biograph 16 scanner was used for a routine scanning of the head to obtain PET images. BrainLAB’s iPlan software was used to fuse 3D T1 images with PET images to obtain PET/MRI images. Results: Focal cortical dysplasia was divided into three types according to ILAE: three patients were classified as type I, twenty-five patients as type II, and forty-one patients as type III. Patients age of onset under 18 and age of operation over 18 had a longer duration (*p* = 0.036, *p* = 0.021). MRI had a high lesion detection sensitivity of type III focal cortical dysplasia (*p* = 0.003). Simultaneous Hybrid PET/MRI showed high sensitivity in detecting type II and III focal cortical dysplasia lesions (*p* = 0.037). The lesions in Simultaneous Hybrid PET/MRI-positive focal cortical dysplasia patients were mostly located in the temporal and multilobar (*p* = 0.005, 0.040). Conclusion: Simultaneous Hybrid PET/MRI has a high accuracy in detecting the classification of focal cortical dysplasia. The results of this study indicate that patients with focal cortical dysplasia with positive Simultaneous Hybrid PET/MRI have better postoperative prognoses.

## 1. Introduction

To date, refractory epilepsy is still a global medical problem, of which approximately 46.5% of patients have focal cortical dysplasia (FCD) [1,2]. FCD is a kind of cortical developmental malformation generated by cortical neuron migration or cell proliferation disorder [3]. In 2011, the International League Against Epilepsy (ILAE) classified focal cortical dysplasia into three subtypes based on microscopic neocortical structural abnormalities [4]: (1) FCD type I is a pathologically focal cortical dysplasia with abnormal longitudinal and/or transverse stratification, in which type I a refers only to structural abnormalities of the cortex, while type I b refers to an abundance of large or immature neurons on the basis of FCD type I a, and the existence of these two changes is called type I c; (2) FCD type II is a heterogenous neuron with or without balloon-like cells, in which type II a is a malformed neuron without balloon cells, while type II b refers to the appearance of balloon-like cells on the basis of FCD type II a; and (3) FCD type III is not only an abnormal stratification of the cortex, but is also accompanied by other lesions such as hippocampal sclerosis and vascular malformation, in which type III a is with hippocampal sclerosis, III b with epileptic-associated tumor, III c with vascular malformation, and III d with early acquired epileptic lesions.

Magnetic resonance imaging (MRI) is the most sensitive form of examination for FCD, and fluid-attenuated inversion recovery (FLAIR) is the most sensitive MRI sequence showing focal cortical thickening, blurred gray matter margins, abnormal gyri morphology, white matter atrophy, and cortical dysplasia, often tapered toward the ventricle [5]. However, the MRI of some patients with FCD were normal [5,6]. Positron emission tomography (PET) imaging can increase the positive rate of focus in MRI-negative FCD patients, and the hypometabolic zone on PET images is generally related to the scope of epileptogenic foci [7,8]. However, due to the lack of clear localization of PET images, the epileptogenic foci cannot be accurately located, so it is impossible to determine whether the hypometabolic zone is located in the cerebral cortex and whether it is the epileptogenic foci [9]. PET/CT is widely used in the preoperative localization of FCD patients [10]. However, the detection rate of lesions in PET/CT imaging is lower than that in PET/MRI imaging [11]. In addition, the radiation dose of PET/CT is much higher than that of Simultaneous Hybrid PET/MRI imaging [12]. Studies have confirmed that compared with either MRI or PET imaging, PET/MRI imaging can improve the sensitivity of epileptic focus detection but cannot improve the specificity, yet PET/MRI imaging is also able to improve the accuracy of epileptic focus as well as reduce bias [9,13].

However, in China, while MRI and PET machines are available in most hospitals, PET/MRI machines are not universal. The hybrid of PET and MRI images can show the localization and characterization of lesions, thus providing richer information to improve the accuracy of lesions in sufferers of FCD [8,13]. Therefore, in the absence of a PET/MRI machine, MRI and PET images can be hybridized with software to improve the detection of FCD lesions. However, the relationship between the Simultaneous Hybrid PET/MRI images of FCD patients and their clinical data, histological classification, and postoperative efficacy has not been much discussed. Therefore, this study mainly investigated the consistency of different histopathological subtypes and the postoperative efficacy of Simultaneous Hybrid PET/MRI images compared with the MRI of FCD. In addition, this study also discussed whether FCD patients have common characteristics in terms of histopathological typing, population statistics variables, and focus location. This study mainly analyzed the general clinical data of the subjects and the preoperative and postoperative characteristics of patients with focal dysplasia on MRI and Simultaneous Hybrid PET/MRI. Finally, the diagnostic and postoperative efficacy of Simultaneous Hybrid PET/MRI for focal dysplasia were evaluated.

## 2. Materials and Methods

### 2.1. Research Plan and Demographic Characteristics

The study was approved by the Ethics Committee. All subjects in this study signed an informed consent. This research retrospectively analyzed 69 patients with FCD who received operative treatments at the First Affiliated Hospital of Soochow University from 2014 to 2020. All patients with FCD were diagnosed histopathologically and underwent a standard preoperative assessment, including electroencephalogram, MRI, and PET scanning. Patients with FCD who could identify the epileptogenic area were treated with surgery, and patients who could not be identified by imaging and other noninvasive tests were treated with intracranial electrode implantation to determine the location of the lesion (Figure 1 and Figure 2). This research is a retrospective analysis of FCD patient data.

### 2.2. Inclusion and Exclusion Criteria of FCD Patients

This research covered surgically treatable diagnosed FCD by the International Alliance Against Epilepsy (ILAE) classification. The exclusion criteria were as follows: (1) the use of EEG data; (2) patients with MRI; (3) patients with large MRI imaging artifacts; and (4) patients with large PET imaging thickness and poor images quality.

### 2.3. Assessment of Clinical Data of FCD Patients

In this study, the year of the first attack, duration of FCD, age of operation and postoperative efficacy of FCD patients were collected and analyzed, and the effect of postoperative epilepsy control in FCD patients was evaluated according to the Engel grading method. The main methods to assess postoperative outcomes are face-to-face interviews or telephone contact with patients.

### 2.4. Histopathological Assessment of Patients with FCD

After surgery, tissue samples from FCD patients were collected in successive sections and stained with conventional hematoxylin and luxol blue. Immunohistochemical analysis was performed using glial fibrillary acidic protein (GFAP) and neuron-specific nuclear protein (NeuN). FCD typing is classified using ILAE criteria. In addition, all patients with FCD underwent EGG examinations as a portion of their preoperative assessment; intracephalic electrode implantation was performed in some patients because imaging of epileptogenic foci could not be determined.

### 2.5. Obtainment and Assessment of the Simultaneous Hybrid PET/MRI and MRI Images

The FCD patients in this study underwent 3.0 Tesla cranial MRI scans that included three-dimensional T1-weighted and T2-fluid-attenuated inversion recovery sequences. Patients underwent PET and MRI imaging within one month of onset. Then, in this study, head MRI was performed using a 3.0 T magnetic resonance scanner (Philips) to obtain 3D T1WI images. The Siemens Biograph 16 scanner was used for a routine scanning of the head to obtain PET images. The scan sequence consisted plain scanned and enhanced 3D T1WI (TE 3. 9 ms, TR 8. 4 ms, matrix 256 × 256, field of view 256 mm × 256 mm, layer thickness 1 mm); 3D T2WI (TE 265 ms, TR 2 500 ms, matrix 256 × 256, field of view 256 mm × 256 mm, layer thickness 1 mm); and 3D T2 FLAIR (TE 325 ms, TR 4 800 ms, Matrix 256 × 256, Field of View 256 mm × 256 mm, Layer thickness 1 mm). Routine PET (Siemens Biograph 16) scanning was performed. The tracer in the examination was 18 F-FDG, fasting was about 6 h before imaging, the patient was instructed to rest quietly before injection, the audiovisual occlusion was about 18 min, the glucose control was less than 11.1 mmol/L, the FDG was 185–370 MBq, and the audiovisual occlusion was continued for 10 min. After resting for 40 min, the imaging began. PET was 3D collection, 1 bed was collected every 5 min, and the thickness of image reconstruction layer was set at 5 mm. BrainLAB’s iPlan software was used to fuse 3D T1 images with PET images to produce PET/MRI images.

Two radiologists evaluated cranial MRI and Simultaneous Hybrid PET/MRI findings to determine the location of the epileptogenic foci. When there is a discrepancy between the two doctors’ assessments, the results are evaluated by a chief imaging physician. Diagnostic criteria: MRI structural abnormalities showing a thickening or thinning of local cerebral cortex; an indistinct boundary of gray matter; cortical or subcortical abnormal signals; lobe atrophy or sulci widening; and hippocampal or temporal lobe volume reduction. The asymmetry index (AI) was used to evaluate suspected epileptic foci in PET image evaluation. AI = (SUV lesion ROi-opposite side ROI of the same layer of SUV)/(SUV lesion ROI+ opposite side ROI of the same layer of SUV). When the absolute value of AI was greater than 15%, it was defined as abnormal, and the anatomical details of MRI fusion images and abnormal signals were evaluated. The physician evaluating the patient’s information is unaware of the patient’s clinical history and EEG results.

When MRI was positive, lesions were observed; when MRI was negative, no lesions were observed. Analogously, Simultaneous Hybrid PET/MRI scans were said to be positive when hypometabolism or cortical structural abnormalities was shown, and negative when hypometabolism and cortical structural abnormalities were not shown. Lesions established on MRI and Simultaneous Hybrid PET/MRI scans were classified, depending on their location, into the following categories: frontal, temporal, parietooccipital, or multilobed.

### 2.6. Statistical Analysis

SPSS 26.0 software was used for the statistical analysis of the data. The mean ± standard deviation (SD), frequency, and percentage were used to display the numerical data. In this study, independent sample t-test was used to compare the measurement data and chi-square test was used to compare the count data between groups [14]. Multiple groups of continuous variables conforming to normal distribution were analyzed by variance. The statistically significant difference was *p* < 0.05. The consistency of the two imaging doctors’ evaluation results was expressed by kappa value. The values of 0.0~0.20 indicated very low consistency, 0.21~0.40 average consistency, 0.41~0.60 medium consistency, 0.61~0.80 high consistency, and 0.81~1 almost exactly.

## 3. Results

### 3.1. Assessment of Population Data of Patients with FCD

There were 69 patients in this study, consisting 30 females and 39 males. According to histopathological examination, FCD patients were divided into three types: three cases (4.35%), twenty-five cases (36.23%), and forty-one cases (59.42%), of which twenty-five cases were located in the temporal region, thirteen cases in the frontal region, eight cases in the parietal occipital region, and twenty-three cases in the multilobular region. Table 1 summarizes the demographic features and findings of the cases. In addition, in the study, the kappa value of consistency assessed by two imaging doctors was 0.75.

Based on the age of first seizure, the mean course of disease in FCD patients younger than 18 years old was 9.60 years (n = 44) and that in FCD patients older than 18 years old was 7.45 years (n = 25). The pathological classification of FCD in the first group was 7%, 43%, and 50%, respectively. In this group: 81.82% of Simultaneous Hybrid PET/MRI was positive, 18.18% of Simultaneous Hybrid PET/MRI was negative, 54.55% of MRI was positive, and 45.45% of MRI was negative (Table 2). In the second group: 24% of type II FCD and 76% of type III FCD were found; 88% of patients of Simultaneous Hybrid PET/MRI were positive, while 12% of patients of Simultaneous Hybrid PET/MRI were negative; and 76% of patients of MRI were positive, while 24% of patients of MRI were negative (Table 2).

According to the age at which FCD patients underwent surgery, 22 patients underwent surgery before 18 years of age, and the mean duration of seizures was 5.83 years, while 47 patients underwent surgery after 18 years of age, and the mean duration of seizures was 10.23 years. The pathological classification of FCD in the first group was 4% for type I, 41% for type II, and 55% for type III. In this group, 95.46% of Simultaneous Hybrid PET/MRI was positive, 4.54% of Simultaneous Hybrid PET/MRI was negative, 68.18% of MRI was positive, and 31.82% of MRI was negative. In the second group, 4% of type I FCD, 34% of type II FCD, and 62% of type III FCD were found; 80.86% of patients of Simultaneous Hybrid PET/MRI were positive, while 19.14% of patients of Simultaneous Hybrid PET/MRI were negative; and 59.57% of patients of MRI were positive, while 40.43% of patients of MRI were negative. There were no statistically significant differences in preoperative (18F)-FDG PET/MRI or MRI results and postoperative pathological results by age at first seizure and age at surgery (Table 2).

According to the statistical analysis of the rank-sum test, FCD cases of those under 18 years old for their first seizure had a longer duration, and of those who were over 18 years old for surgery had a longer duration (*p* = 0.036, *p* = 0.021).

### 3.2. Comparison of MRI Images of FCD Patients

In this research, 43 cases (62.32%) with FCD showed positive MRI scans, 13 of which were found on the temporal pole, 10 of which were located in the frontal lobe, 3 of which were located in the parietal lobe, and 17 of which were located in multiple lobes. In almost all FCD histopathological groups, lesions were found to most often reside on the temporal pole, but FCD type III was more likely to be located in multilobars, although there was no significant difference (*p* = 0.135).

Twenty-six cases were MRI negative, and they were recorded by intracranial electrode implantation, consisting twelve in the temporal region, three in the frontal region, five in the parietal occipital region, and six in the multilobar region. These sites were confirmed postoperatively by histopathology in two cases, fifteen cases, and nine cases. Table 3 shows the pathologic classification and focus localization in the MRI findings of the FCD cases.

Statistical analysis by the chi-square test showed that most of the patients with positive MRI images had type III FCD, while most of the patients with negative MRI images had type I and II FCD (*p* = 0.003).

### 3.3. Comparison of FCD Patients with Preoperative Simultaneous Hybrid PET/MRI Findings

Simultaneous Hybrid PET/MRI images were positive in 59 cases (85.51%), consisting 21 in the temporal region, 10 in the frontal region, 6 in the parietal occipital region, and 22 in the multilobar region. Among the 59 Simultaneous Hybrid PET/MRI-positive FCD patients, there were 23 cases, 35 cases, and 1 case by pathologic classification. There were ten negative Simultaneous Hybrid PET/MRI patients, consisting two, two, and six patients by pathologic classification.

Among the three patients with type I FCD, one was Simultaneous Hybrid PET/MRI positive (33.33%); among the 25 patients with Type II FCD, 23 were Simultaneous Hybrid PET/MRI positive (92%); and among the 41 patients with type III FCD, 35 patients were Simultaneous Hybrid PET/MRI positive (85%). Table 3 shows the comparison of patients with Simultaneous Hybrid PET/MRI findings.

Statistical analysis by the chi-square test showed that the Simultaneous Hybrid PET/MRI findings of focus in cases with type II and III FCD were mostly positive, while Simultaneous Hybrid PET/MRI findings of focus in cases with type I FCD were mostly negative (*p* = 0.037).

### 3.4. FCD Pathological Typing Analyses of MRI and Simultaneous Hybrid PET/MRI Findings

In this research, MRI and Simultaneous Hybrid PET/MRI findings were positive in 43 cases, temporal region in 13 cases, frontal region in 10 cases, parietal occipital region in 3 cases, and multilobar region in 17 cases. According to the histological classification of FCD, there were 1 case, 10 cases, and 32 cases, respectively. Simultaneous Hybrid PET/MRI was positive when MRI findings were positive in FCD patients. There were 16 cases with negative MRI and positive Simultaneous Hybrid PET/MRI. There were eight cases of temporal lesions, three cases of parietal occipital lesions, five cases of multilobar lesions, 81.25% of type II FCD lesions, and 18.75% of type III FCD lesions (Table 4). MRI-negative/Simultaneous Hybrid PET/MRI-negative was found in ten cases; among them, there were four in the temporal region, three in the frontal region, two in the parietal occipital region, and one in the multilobar region, of which type I FCD accounted for 20%, type II FCD accounted for 20%, and type III FCD accounted for 60%.

After comparing the detection rate of lesions with MRI and PET/MRI, the results showed that the histopathological classification and location of lesions of FCD were statistically different between the two groups, while the demographic difference between the two groups was not statistically significant. The histopathological subgroup classification of lesion sites based on MRI and Simultaneous Hybrid PET/MRI scans and the results of postoperative seizures are shown in Table 4 and Table 5. The results of epileptic seizures were good in each group.

Each group had a good postoperative outcome. Among the 43 FCD cases with positive MRI manifestations, 39 cases had a good prognosis and 4 cases had a poor prognosis. Among the 59 FCD cases with positive Simultaneous Hybrid PET/MRI manifestations, 55 cases had a good prognosis and 4 cases had a poor prognosis. Negative MRI and Simultaneous Hybrid PET/MRI were found ten patients, of whom six had a good prognosis and four had a poor prognosis (Table 5).

## 4. Discussion

Several studies have shown relationships between pathologic classification and operative results undergoing MRI or PET examination with FCD [7,15,16]. However, the relationship between operative results in FCD undergoing MRI and Simultaneous Hybrid PET/MRI examination has not been evaluated. Therefore, in comparison with MRI, this research assessed the preoperative Simultaneous Hybrid PET/MRI images and postoperative outcomes of 69 patients.

### 4.1. Assessment of Clinical Data

Cortical dysplasia is a group of diseases composing FCD, which is one of the relatively familiar reasons for pediatric epilepsy surgery [17]. Data from the European Epilepsy Brain Bank estimate that cortical dysplasia accounts for 19.8% of epileptic surgical specimens, of which 70.6% correspond to different FCD subtypes [18]. In one study, FCD had younger onset ages and more frequent episodes than those with other causes [7]. In this research, patients with FCD who were under 18 years old at onset and over 18 years old at surgery had a longer course of disease (Table 2). However, this study also included children and adults and found no statistically significant differences in preoperative Simultaneous Hybrid PET/MRI or MRI results between groups classified by age of seizure.

Type I FCD is most commonly located in the temporal neocortex, specifically the temporal lobe [19]. The age of onset of epilepsy in cases with type I FCD is later, the age requiring surgical treatment is older, and the incidence is lower [20,21]. Type II FCD usually results from damage to the extratemporal region and dysplasia of the multilobar and hemispheric cortices in the frontal lobe [22]. Patients with type II FCD have seizures in childhood that require early and more frequent surgery [23,24,25]. Type III FCD lesions are mainly located in the temporal pole but may also be located in the external temporal position [20]. Type III FCD usually has an adolescent onset of epilepsy [1]. For patients with type III FCD, most drug-resistant epilepsy can be controlled by the surgical removal of the epileptogenic foci [24]. In one study, the majority of type III FCD were reported to be over 18 years old at operation, possibly because of an acquired disease of type III FCD [26]. In this research, patients who underwent surgery after age 18 were also predominantly FCD type III (Table 2).

For example, in the pediatric series at the Freiburg Epilepsy Center, younger patients underwent extensive surgery (i.e., multilobar resection and hemispherectomy) more often than older patients [27]. In addition, patients with multilobar or hemispheric FCD had a younger age of onset and surgery and a shorter duration of onset [28,29,30]. These studies suggest that the histopathological type is a factor causing early epilepticus insultus [30]. In this research, this question was also explored. However, there was no significant difference.

### 4.2. Assessment of Preoperative MRI and Simultaneous Hybrid PET/MRI Findings

MRI images cannot show all FCD lesions. MRI images in 15–40% of epileptic patients are normal [5,31]. There is an agreement that the type of FCD determines whether the lesion can be seen on MRI: MRI can show type II lesions as long as there is a sufficiently clear image and postprocessing is completed [3]. One study found that automatic machine learning analysis of surface-based features could provide a quantitative and objective diagnosis of type II FCD lesions during preoperative evaluation and improve postoperative outcomes [32]. In contrast, cortical cell density in FCD I patients changed only when the tissue was disordered, which was difficult to detect on MRI [20]. According to the Milano group, only half of the patients with type I FCD observed typical FCD changes on MRI [17]. Essentially, 92.80% of patients with type III FCD showed definite lesions on MRI [26]. In this study, MRI scans showed that 2.32% of patients had type I FCD, 23.26% of whom were type II, and 74.42% of whom were type III. Although MRI scans showed better results for type III FCD lesions than for type II and I FCD lesions, there was no significant difference. In MRI-negative patients, FCD was 7.69%, 57.69%, and 34.62% by pathologic classification, respectively. Upon reviewing the literature, postoperative pathological examinations of most MRI images of type I FCD were normal (41–78%), while type II (15–30%) and type III (8–30%) were rare [3,7,26,33]. In this study, MRI-negative patients presented different subgroup distributions. The poor sample capacity may have resulted in this result, with only three people having type I FCD.

In this study, FCD focus was mostly situated on the temporal pole and multilobe in MRI images. Significant differences were seen in histopathological classification, but there were no significant differences in lesion site or demographic variables. In this study, the MRI positive rate of type III FCD was 78.05%, among which type II accounted for 66.67% and Type I accounted for 33.33%. Because of the poor sample capacity of type I FCD, a detailed analysis is not made here. The type III FCD results were similar to previous studies and the type II FCD results were slightly lower than previous studies [3,26]. This result of type II FCD may be related to the poor sample capacity. This question needs to be explored further. The eclamptic zone in MRI-negative patients is difficult to define, and more information can be obtained by Simultaneous Hybrid PET/MRI images as an alternative descriptor in this regard. One study showed that PET/MRI had a sensitivity of 95.3%, specificity of 8.8%, and accuracy of 65.3% in locating seizures in FCD patients based on surgical pathology and postoperative results [34]. Similarly, it was reported that approximately 82.20% (60/73) of patients with FCD PET/MRI scans were positive [35].

In this study, 85.51% of patients were Simultaneous Hybrid PET/MRI positive, of which 59.32% were type III FCD, 38.98% were type II, and 1.69% were type I. Of these patients, the lesion sites were temporal in 21 cases, frontal in 10 cases, parietal occipital in 6 cases, and multilobar in 22 cases. For the demographic characteristics of patients with Simultaneous Hybrid PET/MRI images, cases with (18F)-FDG PET/MRI positivity had earlier seizure age, surgical age, and disease course, but there were no significant differences (Table 3). However, one study showed that PET/MRI images were more likely to show focal epileptic foci in neonates and infants than MRI images [15]. Even more study is needed in this area. In this study, Simultaneous Hybrid PET/MRI showed high sensitivity in the detection of type II and III FCD lesions (*p* = 0.037). The result was similar to previous studies [13,34,36].

### 4.3. Assessment of MRI and Simultaneous Hybrid PET/MRI Findings by FCD Pathological Classification

In this study, preoperative assessment of FCD patients showed positive MRI/ Simultaneous Hybrid PET/MRI in 43 patients. This study found that the accuracy of Simultaneous Hybrid PET/MRI in the detection of focal dysplasia was significantly improved. In particular, the accuracy of foci, temporal lobe, and multilobar foci of type II FCD were higher than that of MRI. Previous studies have shown that the sensitivity increased from 30.1% of patients who showed structural abnormalities on MRI to 94.0% based on simultaneous PET/MRI co-registration [35]. In addition, 53.50% of FCD patients with normal MRI scans PET/MRI revealed cortical hypometabolism [34]; additionally, combined imaging of PET and MRI images should be used as a routine examination method for preoperative assessment because of the conventional negative MRI of FCD patients [37]. The results of these studies report that PET/MRI may well be a helpful preoperative technique for evaluating patients with refractory epilepsy, especially for patients with normal or suspicious MRI results.

In this study, there were 26 MRI-negative patients. Simultaneous Hybrid PET/MRI is positive in all MRI-positive patients because PET/MRI allows a simultaneous assessment of anatomical and functional information [11]. Of the ten negative MRI and PET/MRI, study found that the histopathological classification and location of lesions of FCD were statistically different between the two groups; however, demographics of FCD were not significantly different by statistical analysis. There is little discussion of this issue in the existing literature. This study also assessed visible lesions on MRI and Simultaneous Hybrid PET/MRI. However, the relationship between FDG uptake and lesion size in Simultaneous Hybrid PET/MRI scans was not evaluated, which was also a limitation of this study.

This study showed that FCD lesions in the temporal pole were mostly of negative MRI. However, there was no statistically significant difference. This study suggests that most FCD patients after surgery have good seizure control. Moreover, patients with positive PET/MRI results had a good prognosis. This study showed that 93% of FCD patients with positive PET/MRI had a good prognosis, while 90% of FCD patients with positive MRI had a good prognosis. Therefore, (18F)-FDG PET and MRI combined imaging can not only help the noninvasive localization of epileptic foci, but also improve the surgical efficacy of patients with FCD. This result was similar to previous studies [34].

Finally, there are some limitations to this study. (18F)-FDG PET/MRI combination imaging is performed on images acquired on different machines and at different times. Therefore, errors may be caused by incorrect registration or various motion artifacts during brain scan.

## 5. Conclusions

In conclusion, the hybrid imaging of PET/MRI can help identify epileptic lesions. Epileptic lesions with positive imaging manifestations should be controlled effectively for a long time by surgery. However, despite the continuous innovation of imaging technology, not all types of FCD can be displayed, thus we still need to further discuss the detection of FCD diseases.

## Figures and Tables

**Figure 1 brainsci-13-00611-f001:**
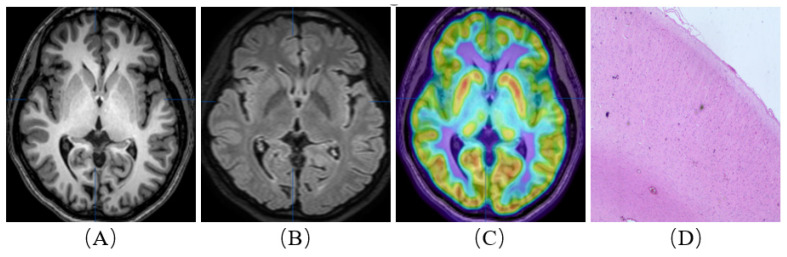
A patient with refractory epilepsy to drug therapy. She was born in 1993 and was admitted when she was 26 years old. Her first seizure occurred at 16 years old. Neither MRI nor (18F)-FDG PET/MRI (**A**–**C**) showed the epileptic focus. Resective surgery was planned for the patient, and an intracranial electrode was used. The intracranial electrode indicated the right frontal lobe region. Postoperative histopathological examination confirmed ‘type Ⅰ b, focal cortical dysplasia’ (**D**). The patient is currently seizure-free (Engel III–IV).

**Figure 2 brainsci-13-00611-f002:**
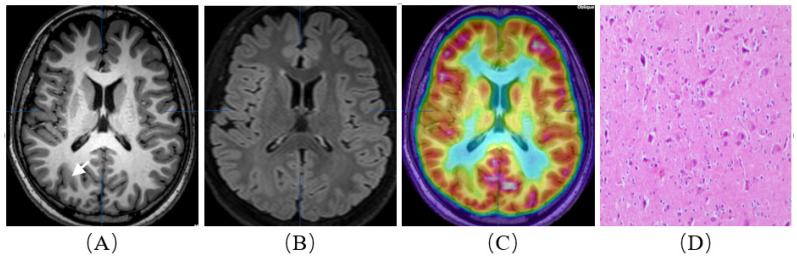
A patient with refractory epilepsy to drug therapy. He was born in 2003 and was admitted when he was 14 years old. His first seizure occurred at two years old. Interictal and ictal EEG recordings indicated the right parietal occipital lobe region. MRI showed cortical and subcortical signal abnormalities on axial TIWI (**A**) and FLAIR T2WI (**B**) in the right parietal occipital lobe. (18F)-FDG PET/MRI images (**C**) demonstrated hypometabolism in the parietal occipital lobe extending also to the parietal occipital area on the right. Resective surgery was planned for the patient, and an intracranial electrode was not used. Postoperative histopathological examination (**D**) confirmed ‘type II a, focal cortical dysplasia’. The patient is currently seizure-free (Engel I).

**Table 1 brainsci-13-00611-t001:** Demographic characteristics of patients, (ASO: age of seizure onset).

Projects		Gender			Classification of FCD			
	All FCD Patients (n = 69)	Female (n = 30)	Male (n = 39)	*p* Value	Type Ⅰ FCD (n = 3)	Type Ⅱ FCD (n = 25)	Type Ⅲ FCD (n = 41)	*p* Value
ASO	16.64 ± 12.86	15.93 ± 13.65	17.18 ± 12.36	0.693	9.67 ± 6.51	12.04 ± 8.87	19.95 ± 12.27	0.058
Duration of epilepsy	8.83 ± 7.03	10.28 ± 7.45	7.71 ± 6.58	0.128	13.33 ± 5.77	9.80 ± 7.72	7.90 ± 6.61	0.221
Age at surgery	25.46 ± 12.96	26.21 ± 13.88	24.89 ± 12.35	0.677	23.00 ± 8.89	21.84 ± 10.10	27.85 ± 14.35	0.252

**Table 2 brainsci-13-00611-t002:** Demographic data of patients according to age at surgery and seizure onset.

	Age of Seizure Onset			Age at Surgery		
	<18 years (n = 44)	>18 years (n = 25)	*p* value	<18 years (n = 22)	>18 years (n = 47)	*p* value
ASO	8.76 (0–18)	30.50 (19–55)	-	5.72 (0–14)	21.75 (1–55)	-
Duration of epilepsy	9.60 (0.7–28)	7.45 (0.02–23)	0.036	5.83 (0.7–14)	10.23 (0.02–28)	0.021
Age at surgery	18.36 (5–36)	37.96 (22–62)	-	11.55 (6–17)	31.98 (19–62)	-
FCD type			0.106			0.817
Type I FCD	3 (7%)	0		1 (4%)	2 (4%)	
Type II FCD	19 (43%)	6 (24%)		9 (41%)	16 (34%)	
Type III FCD	22 (50%)	19 (76%)		12 (55%)	29 (62%)	
MRI findings			0.064			0.340
MRI-positive	24 (54.55%)	19 (76.00%)		15 (68.18%)	28 (59.57%)	
MRI-negative	20 (45.45%)	6 (24.00%)		7 (31.82%)	19 (40.43%)	
Simultaneous Hybrid PET/MRI findings			0.734			0.103
Simultaneous Hybrid PET/MRI-positive	36 (81.82%)	22 (88.00%)		21 (95.46%)	38 (80.86%)	
Simultaneous Hybrid PET/MRI-negative	8 (18.18%)	3 (12.00%)		1 (4.54%)	9 (19.14%)	

**Table 3 brainsci-13-00611-t003:** Comparison of MRI-positive, MRI-negative, Simultaneous Hybrid PET/MRI-positive, and Simultaneous Hybrid PET/MRI-negative patients.

	MRI-Positive (n = 43)	MRI-Negative (n = 26)	*p* Value	Simultaneous Hybrid PET/MRI-Positive (n = 59)	Simultaneous Hybrid PET/MRI-Negative (n = 10)	*p* Value
FCD type			0.003			0.037
Type I FCD	1 (2%)	2 (8%)		1 (2%)	2 (20%)	
Type II FCD	10 (23%)	15 (58%)		23 (39%)	2 (20%)	
Type III FCD	32 (74%)	9 (34%)		35 (59%)	6 (60%)	
FCD location			0.135			0.073
Temporal	13 (30%)	12 (47%)		21 (36%)	4 (40%)	
Frontal	10 (23%)	3 (13%)		10 (17%)	3 (30%)	
Parietooccipital	3 (7%)	5 (19%)		6 (10%)	2 (20%)	
Multilobar	17 (40%)	6 (21%)		22 (37%)	1 (10%)	
Age at onset	17.97 ± 13.20	14.44 ± 12.19	0.258	16.34 ± 12.53	18.40 ± 15.25	0.696
Duration of epilepsy	7.73 ± 6.16	10.64 ± 8.08	0.186	8.41 ± 6.78	11.30 ± 8.31	0.320
Age at surgery	25.70 ± 13.44	25.08 ± 12.36	0.489	24.74 ± 12.79	29.70 ± 13.81	0.378

**Table 4 brainsci-13-00611-t004:** MRI-positive/Simultaneous Hybrid PET/MRI-positive, MRI-positive/Simultaneous Hybrid PET/MRI-negative, MRI-negative/Simultaneous Hybrid PET/MRI-positive, and MRI-negative/Simultaneous Hybrid PET/MRI-negative patients.

	MRI-Positive/Simultaneous Hybrid PET/MRI-Positive (n = 43)	MRI-Positive/Simultaneous Hybrid PET/MRI-Negative (n = 0)	MRI-Negative/Simultaneous Hybrid PET/MRI-Positive (n = 16)	MRI-Negative/Simultaneous Hybrid PET/MRI-Negative (n = 10)	*p* Value
Type I	1 (2%)	-	-	2 (20%)	1.000
Type II	10 (23%)	-	13 (81%)	2 (20%)	0.000
Type III	32 (74%)	-	3 (19%)	6 (60%)	0.391
Temporal	13 (30%)	-	8 (50%)	4 (40%)	0.005
Frontal	10 (23%)	-	-	3 (30%)	1.000
Parietooccipital	3 (7%)	-	3 (19%)	2 (20%)	0.131
Multilobar	17 (40%)	-	5 (31%)	1 (10%)	0.040
Age at onset	17.97 ± 13.20	-	11.96 ± 9.54	18.40 ± 15.25	0.254
Duration of epilepsy	7.73 ± 6.16	-	10.23 ± 8.17	11.30 ± 8.31	0.236
Age at surgery	25.70 ± 13.41	-	22.19 ± 10.82	29.70 ± 13.81	0.347

**Table 5 brainsci-13-00611-t005:** Comparison of four groups (MRI+/Simultaneous Hybrid PET/MRI+, MRI+/Simultaneous Hybrid PET/MRI−, MRI-/Simultaneous Hybrid PET/MRI+, MRI−/Simultaneous Hybrid PET/MRI−) based on outcomes.

	Good Outcome		Poor Outcome
	Engel I	Engel II	Engel III + IV
MRI+/Simultaneous Hybrid PET/MRI+ (n = 43)	30	9	4
MRI+/Simultaneous Hybrid PET/MRI− (n = 0)	-	-	-
MRI−/Simultaneous Hybrid PET/MRI+ (n = 16)	13	3	-
MRI−/Simultaneous Hybrid PET/MRI− (n = 10)	3	3	4

## Data Availability

Not applicable.

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
