# Peer review of "An Assessment of the Pathological Classification and Postoperative Outcome of Focal Cortical Dysplasia by Simultaneous Hybrid PET/MRI"

_brainsci, 2023, doi:10.3390/brainsci13040611_

Round 1
Reviewer 1 Report
-Please introduce the abbreviations of MRI, CT and FLAIR.
-Authors use TIWI only once and does not need to abbreviate it. please use the complete format.
-After introducing authors contribution in section introduction, it is recommended to present how the paper is organized and what are the next steps.
-Please mention that you had the consent letters from the candidates who participated in the project.
-Please mention if the study has ethical committee letter/number.
-authors employed ‘’the chi-square 169 test, T test and rank sum test’’ for statistical analysis. please provide a reference for the methods.
-Please polish the columns topics in table 1 by editorial modifications.
-The study is application based and investigating a comparison between predefined software and using hybrid technology methods for identifying epilepsy. I wish authors also consider the mathematical methods which are employed in the technologies to address the deficits or limitations of the medical devices and why the hybrid method has more significant results.
Author Response
-Please introduce the abbreviations of MRI, CT and FLAIR.
fluid-attenuated inversion recovery (FLAIR)
Answer: The article has been revised according to your suggestion. Modify the content: Magnetic resonance imaging (MRI), computed tomography (CT), fluid-attenuated inversion recovery (FLAIR).
-Authors use TIWI only once and does not need to abbreviate it. please use the complete format.
Answer: The article has been revised according to your suggestion. Modify the content: T1-weighted imaging.
-After introducing authors contribution in section introduction, it is recommended to present how the paper is organized and what are the next steps.
Answer: The article has been revised according to your suggestion. Modify the content: This study mainly analyzed the general clinical data of the subjects, and analyzed the preoperative and postoperative characteristics of patients with focal dysplasia on MRI and hybrid PET/MRI. Finally, the diagnostic and postoperative efficacy of hybrid PET/MRI for focal dysplasia were evaluated.
-Please mention that you had the consent letters from the candidates who participated in the project.
Answer: The article has been revised according to your suggestion. Modify the content: All subjects in this study signed informed consent.
-Please mention if the study has ethical committee letter/number.
Answer: The article has been revised according to your suggestion. Modify the content: The study was approved by the Ethics Committee, ethics approval number 137.
-authors employed ‘’the chi-square 169 test, T test and rank sum test’’ for statistical analysis. please provide a reference for the methods.
Answer: The article has been revised according to your suggestion. Modify the content: In this study, independent sample t test was used to compare the measurement data and chi-square test was used to compare the count data between groups [14].
-Please polish the columns topics in table 1 by editorial modifications.
Answer: The article has been revised according to your suggestion. Modify the content:
projects |
|
gender |
classification of FCD |
|||||
|
All FCD patients(n=69) |
Female(n=30) |
Male(n=39) |
P value |
Typeâ… FCD(n=3) |
TypeⅡ FCD(n=25) |
TypeⅢ FCD(n=41) |
P value |
ASO |
16.64±12.86 |
15.93±13.65 |
17.18±12.36 |
0.693 |
9.67±6.51 |
12.04±8.87 |
19.95±12.27 |
0.058 |
Duration of epilepsy |
8.83±7.03 |
10.28±7.45 |
7.71±6.58 |
0.128 |
13.33±5.77 |
9.80±7.72 |
7.90±6.61 |
0.221 |
Age at surgery |
25.46±12.96 |
26.21±13.88 |
24.89±12.35 |
0.677 |
23.00±8.89 |
21.84±10.10 |
27.85±14.35 |
0.252 |
-The study is application based and investigating a comparison between predefined software and using hybrid technology methods for identifying epilepsy. I wish authors also consider the mathematical methods which are employed in the technologies to address the deficits or limitations of the medical devices and why the hybrid method has more significant results.
Answer: The article has been revised according to your suggestion. Modify the content: Due to the limitation of PET spatial resolution and the influence of partial volume effect, the boundary between lesions and surrounding structures will be blurred, the anatomical details will not be clearly displayed, and the actual radiation absorption value of small lesions may be underestimated. Moreover, the spatial resolution will also affect the detection of lesions. PET and MRI technologies can realize synchronous data acquisition and image fusion, so as to obtain comprehensive information of human body structure, function and metabolism, and reduce radiation hazards, which will have important clinical value for personalized diagnosis and treatment of precision medicine. The imaging mode combined with MRI and PET has been widely recognized. The two complement each other. The images obtained have the advantages of high spatial resolution and high contrast of MRI, as well as the characteristics of high sensitivity and molecular level imaging of PET imaging.

Reviewer 2 Report
Overall very nice study and useful for clinical setup and should publish, however, I have the following comments to improve the manuscript and increases the readers' interest.
Authors used the term hybrid PETMR, however, it looks like in this study authors collected MRI and PET data separately and combined them during post-processing, so it would be nice to clarify somewhere in the methodology as readers may understand it as simultaneous hybrid PETMR.
MRI structural abnormalities showed thickening or thinning of the local cerebral cortex; Indistinct boundary of gray matter; Cortical or subcortical abnormal signals: lobe atrophy or sulci widening; Hippocampal or temporal lobe volume reduction: Is this visual assessment if yes it would be great if authors can run some quantitative measurement for cortical thinning using T1-weighted high-resolution scans?
More information about the methodology/imaging parameters of MRI and PET, and also about the data processing of PETMR both in the abstract and also in the methodology section of the main body. Also, it is not very clear how authors measure the PET activation area. Is this quantitative or qualitative?
Statistical analysis: Authors should include an interobserver agreement analysis in the manuscript and it would be nice if authors can report the sensitivity/specificity/ ROC curves along with areas under the ROC (AUCs).
It would be nice to include the age/gender-matched control, however, it is hard to find out, so authors should mention it as a limitation in the study.
What is ILAE in the abstract? Please explain at the first time.
Focal cortical dysplasia was divided into 3 types: 3 cases, 25 cases 23, and 41 cases: please rephrase this line.
Author Response
- Authors used the term hybrid PETMR, however, it looks like in this study authors
collected MRI and PET data separately and combined them during post-processing, so it would be nice to clarify somewhere in the methodology as readers may understand it as simultaneous hybrid PETMR.
Answer: The article has been revised according to your suggestion. Modify the content: In this study, head MRI was performed using a 3.0T magnetic resonance scanner (Philips) to obtain 3D T1WI images. The Siemens Biograph 16 scanner was used for routine scanning of the head to obtain PET images. BrainLAB's iplan software was used to fuse 3D T1 images with PET images to obtain PET/MRI images.
- MRI structural abnormalities showed thickening or thinning of the local cerebral cortex; Indistinct boundary of gray matter; Cortical or subcortical abnormal signals: lobe atrophy or sulci widening; Hippocampal or temporal lobe volume reduction: Is this visual assessment if yes it would be great if authors can run some quantitative measurement for cortical thinning using T1-weighted high-resolution scans?
Answer: The article has been revised according to your suggestion. Modify the content: In this study, signs of reduced volume of the hippocampus or temporal lobe were visually assessed. In this study, if the volume of the patient's hippocampus or temporal lobe changes and the images of such changes are obvious, visual evaluation is used. When the change is not significant, use quantitative evaluation.
- More information about the methodology/imaging parameters of MRI and PET, and also about the data processing of PETMR both in the abstract and also in the methodology section of the main body. Also, it is not very clear how authors measure the PET activation area. Is this quantitative or qualitative?
Answer: The article has been revised according to your suggestion. Modify the content: MRI was performed using 3.0 T magnetic resonance imaging (MRI). The scan sequence consisted of plain scan and enhanced 3D T1WI (TE 3. 9 ms, TR 8. 4 ms, matrix 256 × 256, field of view 256 mm × 256 mm, layer thickness 1 mm); 3D T2WI (TE 265 ms, TR 2 500 ms, matrix 256 × 256, field of view 256 mm × 256 mm, layer thickness 1 mm); 3D T2 FLAIR (TE 325 ms, TR 4 800 ms, Matrix 256 × 256, Field of View 256 mm × 256 mm, Layer thickness 1 mm). Routine PET (Siemens Biograph 16) scanning was performed. The tracer in the examination was 18 F-FDG, fasting was about 6 h before imaging, the patient was instructed to rest quietly before injection, the audiovisual occlusion was about 18 min, the glucose control was less than 11.1 mmol/L, the FDG was 185-370 MBq, and the audiovisual occlusion was continued for 10 min After resting for 40 min, the imaging began. PET was 3D collection, 1 bed was collected every 5 min, and the thickness of image reconstruction layer was set at 5 mm. BrainLAB's iplan software was used to fuse 3D T1 images with PET images to produce PET/MRI images.
The asymmetry index (AI) was used to evaluate suspected epileptic foci in PET image evaluation. AI= (SUV lesion ROi-opposite side ROI of the same layer of SUV)/ (SUV lesion ROI+ opposite side ROI of the same layer of SUV). When the absolute value of AI was greater than 15%, it was defined as abnormal, and the anatomical details of MRI fusion images and abnormal signals were evaluated.
- Statistical analysis: Authors should include an interobserver agreement analysis in the manuscript and it would be nice if authors can report the sensitivity/specificity/ ROC curves along with areas under the ROC (AUCs).
Answer: The article has been revised according to your suggestion. Modify the content: The consistency of the two imaging doctors' evaluation results was expressed by Kappa value. 0.0~0.20 very low consistency, 0.21~0.40 average consistency, 0.41~0.60 medium consistency, 0.61~0.80 high consistency, and 0.81-1 almost exactly. In the study, the kappa value of consistency assessed by two imaging doctors was 0.75.
- It would be nice to include the age/gender-matched control, however, it is hard to find out, so authors should mention it as a limitation in the study.
Answer: The article has been revised according to your suggestion. Modify the content: Finally, there are some limitations to this study. First of all, (18F)-FDG PET/MRI combination imaging is performed on images acquired on different machines and at different times. Therefore, errors may be caused by incorrect registration or various motion artifacts during brain scan. The second is not the age/gender-matched control.
- What is ILAE in the abstract? Please explain at the first time.
Answer: The article has been revised according to your suggestion. Modify the content: ILAE is International League Against Epilepsy.
- Focal cortical dysplasia was divided into 3 types: 3 cases, 25 cases 23, and 41 cases: please rephrase this line.
Answer: The article has been revised according to your suggestion. Modify the content: Focal cortical dysplasia was divided into 3 types according to ILAE: 3 patients were classified as type I, 25 patients as type II, and 41 patients as type III.

Reviewer 3 Report
The purpose of this study was to investigate whether MRI and hybrid PET/MRI images are concordant in the histologic classification of patients with focal cortical dysplasia. In addition, the postoperative outcome of focal cortical dysplasia was assessed by MRI and hybrid PET/MRI images. This article has some value for the detection of focal cortical dysplasia lesions in medicine. The whole article concludes that the positive rate of hybrid PET/MRI is higher for type II and type III focal cortical dysplasia, demonstrated that using hybrid PET/MRI imaging to detect lesions, they could then better control seizures by resecting the lesions. But I have some suggestions for the content and structure of the paper. The authors need to pay attention to the following suggestions and make improvements.
1. The motivation is not clear. Please specify the importance of the proposed solution.
2. Please highlight the contributions/innovations of the proposed solution in introduction.
3. Authors ignore some recently published solutions, such as "Brain tuner segmentation based on the fusion of deep semantics and edge information in multimodal MRI" Information Fusion Volume 91: 376-387, 2023 and "Automatic Detection of Focal Cortical Dysplasia Type II in MRI: Is the Application of Surface-Based Morphometry and Machine Learning Promising?", Front. Hum. Neurosci., Sec. Brain Imaging and Stimulation, 15:608285, 2021. Please discuss them in this paper.
4. In the writing of the proposed materials and methods, the author systematically expounded the relevant information of FCD patients. Acquisition and evaluation of PET-MRI and MRI hybrid images are described.
5. In the introduction of the second part, it is difficult to see the method mentioned in the article. The author refers to "do you use SPSS for statistical analysis?" Moreover, there are few analyzes in this part, so it is difficult to support the conclusion.
6. PET/MRI is a new human detection technology that can better diagnose the health of the human body. Has the author considered applying it to other scenarios, not just the detection and diagnosis of patients with focal cortical dysplasia?
7. In the discussion and analysis of the fourth part, it can be seen that the author's thinking is rigorous and relatively complete, but there is a lack of analysis combined with image detection results.
8. More technical details of classification should be given.
Author Response
The purpose of this study was to investigate whether MRI and hybrid PET/MRI images are concordant in the histologic classification of patients with focal cortical dysplasia. In addition, the postoperative outcome of focal cortical dysplasia was assessed by MRI and hybrid PET/MRI images. This article has some value for the detection of focal cortical dysplasia lesions in medicine. The whole article concludes that the positive rate of hybrid PET/MRI is higher for type II and type III focal cortical dysplasia, demonstrated that using hybrid PET/MRI imaging to detect lesions, they could then better control seizures by resecting the lesions. But I have some suggestions for the content and structure of the paper. The authors need to pay attention to the following suggestions and make improvements.
- The motivation is not clear. Please specify the importance of the proposed solution.
Answer: The article has been revised according to your suggestion. Modify the content: Motivation of this article: PET/MRI imaging can improve the sensitivity and accuracy of epilepsy focus detection and reduce bias. However, in China, most hospitals have magnetic resonance imaging machines, but PET-MRI machines are rare. Therefore, this study evaluated the consistency of different histopathologic subtypes of FCD lesions between hybrid PET/MRI imaging and MRI. Postoperative outcomes were evaluated in patients between the two groups.
- Please highlight the contributions/innovations of the proposed solution in introduction.
Answer: The article has been revised according to your suggestion. Modify the content: 1. This study found that hybrid PET/MRI has a more accurate detection of focal cortical dysplasia. 2. The results of this study indicate that patients with focal cortical dysplasia with positive hybrid PET/MRI have better postoperative prognosis.
- Authors ignore some recently published solutions, such as "Brain tumor segmentation based on the fusion of deep semantics and edge information in multimodal MRI" Information Fusion Volume 91: 376-387, 2023 and "Automatic Detection of Focal Cortical Dysplasia Type II in MRI: Is the Application of Surface-Based Morphometry and Machine Learning Promising?", Front. Hum. Neurosci., Sec. Brain Imaging and Stimulation, 15:608285, 2021. Please discuss them in this paper.
Answer: The article has been revised according to your suggestion. Modify the content: One study found that automatic machine learning analysis of surface-based features could provide a quantitative and objective diagnosis of type II FCDlesions during preoperative evaluation and improve postoperative outcomes[31].
- In the writing of the proposed materials and methods, the author systematically expounded the relevant information of FCD patients. Acquisition and evaluation of PET-MRI and MRI hybrid images are described. In the introduction of the second part, it is difficult to see the method mentioned in the article. The author refers to "do you use SPSS for statistical analysis?" Moreover, there are few analyzes in this part, so it is difficult to support the conclusion.
Answer: The article has been revised according to your suggestion. Modify the content: SPSS 26.0 software was used for statistical analysis of the data. The mean ± standard deviation (SD), frequency and percentage were used to display the numerical data. In this study, independent sample t test was used to compare the measurement data and chi-square test was used to compare the count data between groups. Multiple groups of continuous variables conforming to normal distribution were analyzed by variance. The statistically significant difference was P < 0.05. The consistency of the two imaging doctors' evaluation results was expressed by Kappa value. 0.0~0.20 very low consistency, 0.21~0.40 average consistency, 0.41~0.60 medium consistency, 0.61~0.80 high consistency, and 0.81-1 almost exactly. Therefore, in this study, the P-values obtained from data analysis in Tables 1 to 4 were all obtained through independent sample T-test and Chi-square test analysis of SPSS. As can be seen from the results in Table 4, compared with MRI, the accuracy of hybrid PET/MRI in detecting lesions with focal dysplasia is significantly higher. In addition, the detection accuracy of type II FCD, temporal lobe and multilobar foci was greater than that of MRI.
Table 4 MRI-positive/ hybrid PET/MRI-positive, MRI-positive/ hybrid PET/MRI-negative, MRI-negative/ hybrid PET/MRI-positive, and MRI-negative/ hybrid PET/MRI-negative patients.
projects |
MRI-positive/ hybrid PET/MRI-positive(n=43) |
MRI-positive/ hybrid PET/MRI-negative(n=0) |
MRI-negative/ hybrid PET/MRI-positive(n=16) |
MRI-negative/ hybrid PET/MRI-negative(n=10) |
P value |
FCD |
43(62%) |
- |
16(23%) |
10(15%) |
0.002 |
Typeâ… |
1(2%) |
- |
- |
2(20%) |
1.000 |
Typeâ…¡ |
10(23%) |
- |
13(81%) |
2(20%) |
0.000 |
Typeâ…¢ |
32(74%) |
- |
3(19%) |
6(60%) |
0.391 |
Temporal |
13(30%) |
- |
8(50%) |
4(40%) |
0.005 |
Frontal |
10(23%) |
- |
- |
3(30%) |
1.000 |
Parietooccipital |
3(7%) |
- |
3(19%) |
2(20%) |
0.131 |
Multilobar |
17(40%) |
- |
5(31%) |
1(10%) |
0.040 |
Age at onset |
17.97±13.20 |
- |
11.96±9.54 |
18.40±15.25 |
0.254 |
Duration of epilepsy |
7.73±6.16 |
- |
10.23±8.17 |
11.30±8.31 |
0.236 |
Age at surgery |
25.70±13.41 |
- |
22.19±10.82 |
29.70±13.81 |
0.347 |
- PET/MRI is a new human detection technology that can better diagnose the health of the human body. Has the author considered applying it to other scenarios, not just the detection and diagnosis of patients with focal cortical dysplasia?
Answer: The images obtained by PET/MRI have the advantages of high spatial resolution and high contrast of MRI, as well as the characteristics of high sensitivity and molecular level imaging of PET imaging. Therefore, in addition to the detection and diagnosis of patients with focal cortical dysplasia, we also apply the fusion technology of PET/MRI to the detection and diagnosis of intracranial tumors, Alzheimer's disease and autoimmune diseases.
- In the discussion and analysis of the fourth part, it can be seen that the author's thinking is rigorous and relatively complete, but there is a lack of analysis combined with image detection results.
Answer: The article has been revised according to your suggestion. Modify the content: This study found that the accuracy of hybrid PET/MRI in the detection of focal dysplasia was significantly improved. In particular, the accuracy of foci, temporal lobe and multilobar foci of type II FCD was higher than that of MRI.
This study found that the histopathological classification and location of lesions of FCD were statistically different between the two groups, However, demographics of FCD were not significantly different by statistical analysis.
This study shows that 93% of FCD patients with positive PET/MRI had a good prognosis, while 90% of FCD patients with positive MRI had a good prognosis. Therefore, (18F)-FDG PET and MRI combined imaging can improve the surgical efficacy of patients with FCD.
- More technical details of classification should be given.
Answer: The article has been revised according to your suggestion. Modify the content: MRI was performed using 3.0-t magnetic resonance imaging (MRI). The scan sequence consisted of plain scan and enhanced 3D T1WI; 3D T2WI; 3D T2 FLAIR. Routine PET(Siemens Biograph 16) scanning was performed. BrainLAB's iplan software was used to fuse 3D T1 images with PET images to produce PET/MRI images. When MRI was positive, lesions were observed; when MRI was negative, no lesions were observed. Analogously, hybrid PET/MRI scans were said to be positive when hypometabolism or cortical structural abnormalities was shown and negative when hypometabolism and cortical structural abnormalities were not shown.

Round 2
Reviewer 3 Report
All my concerns have been addressed. I recommend this paper for publication.